# A CT radiomics analysis of COVID-19-related ground-glass opacities and consolidation: Is it valuable in a differential diagnosis with other atypical pneumonias?

**Mutlu Gülbay**[1]*, **Bahadır Orkun Özbay**[2], **Bökebatur Ahmet Raşit Mendi**[1], **Aliye Baştuğ**[2], **Hürrem Bodur**[2]

1 Department of Radiology, Ankara Numune Education and Research Hospital, Ankara City Hospital, Universiteler Mahallesi, Ankara, Çankaya, Turkey, 2 Department of Infectious Diseases and Clinical Microbiology, Ankara Numune Education and Research Hospital, Ankara City Hospital, Universiteler Mahallesi, Ankara, Çankaya, Turkey

* drgulbay@gmail.com

**Data Availability Statement:** All relevant data are within the paper.

## Abstract

### Purpose

To evaluate the discrimination of parenchymal lesions between COVID-19 and other atypical pneumonia (AP) by using only radiomics features.

### Methods

In this retrospective study, 301 pneumonic lesions (150 ground-glass opacity [GGO], 52 crazy paving [CP], 99 consolidation) obtained from nonenhanced thorax CT scans of 74 AP (46 male and 28 female; 48.25±13.67 years) and 60 COVID-19 (39 male and 21 female; 48.01±20.38 years) patients were segmented manually by two independent radiologists, and Location, Size, Shape, and First- and Second-order radiomics features were calculated.

### Results

Multiple parameters showed significant differences between AP and COVID-19-related GGOs and consolidations, although only the Range parameter was significantly different for CPs. Models developed by using the Bayesian information criterion (BIC) for the whole group of GGO and consolidation lesions predicted COVID-19 consolidation and AP GGO lesions with low accuracy (46.1% and 60.8%, respectively). Thus, instead of subjective classification, lesions were reclassified according to their skewness into positive skewness group (PSG, 78 AP and 71 COVID-19 lesions) and negative skewness group (NSG, 56 AP and 44 COVID-19 lesions), and group-specific models were created. The best AUC, accuracy, sensitivity, and specificity were respectively 0.774, 75.8%, 74.6%, and 76.9% among the PSG models and 0.907, 83%, 79.5%, and 85.7% for the NSG models. The best PSG model was also better at predicting NSG lesions smaller than 3 mL. Using an algorithm, 80% of COVID-19 and 81.1% of AP patients were correctly predicted.

**Funding:** The authors received no specific funding for this work.

**Competing interests:** The authors have declared that no competing interests exist.

## Conclusion

During periods of increasing AP, radiomics parameters may provide valuable data for the differential diagnosis of COVID-19.

## Introduction

The disease caused by the SARS-CoV-2 virus that first appeared in Wuhan, China, at the end of 2019 was named COVID-19 by the WHO on February 11, 2020. As of February 5, 2021, there were 104,370,550 confirmed cases of COVID-19, including 2,271,380 deaths, reported to the WHO [1]. The early stages of the COVID-19 pandemic coincided with the cold season in the Northern Hemisphere, when other respiratory infections were also common. In the Expert Consensus Statement on Reporting Chest CT Findings Related to COVID-19, it was recommended to mention other etiologies, such as influenza, in the differential diagnosis, even for a typical COVID-19 pneumonia radiology result [2]. A late diagnosis for influenza has been reported to be associated with an increased likelihood of developing complications and the length of hospital stay [3], especially in the elderly population [4]. Although increased admissions with heart failure were reported in late-diagnosed COVID-19 patients, no significant increases were observed in ICU admissions and mortality rates [5]. Symptoms of respiratory infection and pulmonary infiltrates with negative RT-PCR tests have been reported for COVID-19 [6,7] and atypical pneumonia (AP) [8,9] patients. Sensitivity of RT-PCR for SARS-CoV-2 was reported 89% (95% CI, 81%-94%) while sensitivity of CT for COVID-19 pneumonia was calculated 94,6% (95% CI, 91.9–96,4%) according to the pooled data [10]. Despite the high sensitivity of CT, it has been reported that its specificity was 46% (95% CI, 31.9%-60.7%) only [10]. These data point to the cases with typical or indeterminate CT findings in the presence of false-negative PCR results and is a cause of diagnostic confusion [11]. Thus, radiological methods to assist the decision-making process are of increasing interest [12–14].

Radiomic analysis evaluates the texture, shape and size characteristics of any type of tissue using the voxels in the cross-sectional images obtained from scans [15]. The methods used to achieve said analysis have standardization and reproducibility issues; thus, it is not widely used in daily radiologic work-ups [15,16]. However, an increasing number of publications have reported the use of radiomic parameters mostly in the evaluation of neoplastic lesions of the lung [16–19]. Recently, studies evaluating the pneumonic lesions of COVID-19 were added to these publications. Radiomic parameters have been reported to be effective in assessing the severity of the disease [20], determining the prognosis [21] and differentiating it from other AP [22]. In previous radiomics studies on COVID-19, mixed models consisting of radiomic parameters combined with various clinical and laboratory findings were frequently used. Although many parameters, such as fever, saturation values, liver function tests and blood cell count, are used in these models, they are nonspecific for AP. A lesion-based evaluation can provide valuable information in the evaluating of suspicious cases with a negative PCR test and indeterminate parenchymal lesions.

The aim of this study was to evaluate the ability of CT radiomics parameters and models to discriminate COVID-19 and AP lesions without the use of any other clinical or laboratory data. For this purpose, different models were created and compared using validated results, and the efficiency of an algorithm based on these models alone, instead of combining with other clinical and laboratory data, in classifying COVID-19 and AP lesions was evaluated.

## Materials and methods

This retrospective, cross-sectional study was approved by the institutional review board, and written informed patient consent was waived.

### Study population

**Inclusion criteria.** Ninety-eight consecutive patients who were admitted to our hospital between May 2019 and April 2020 and diagnosed with AP by multiplex RT-PCR panel for Adenovirus, Bocavirus, Coronavirus 229E, Coronavirus HKU1, Coronavirus NL63, Coronavirus OC43, Enterovirus, Human metapneumovirus A/B, Influenza A, Influenza A (H1N1), Influenza B, *Mycoplasma pneumoniae*, Parainfluenza 1, 2, 3 and 4, Parechovirus, Respiratory syncytial virus A/B, Rhinovirus (Fast Track FTD Respiratory Pathogens 21 kit, Fast Track Diagnosis, Luxembourg) or urinary antigen test and 80 consecutive patients diagnosed with COVID-19 by RT-PCR for SARS-CoV-2 (Bio-Speedy COVID-19 qPCR Detection Kit, Cat No: BS-SY-WC-305, Bioeksen, Turkey) in April 2020 were evaluated.

In both AP group and COVID-19 group, patients with a positive viral PCR or urinary antigen test and a nonenhanced thorax CT scan before any antiviral treatment, who had no proven bacterial infection, had no parenchymal findings of other lung diseases, had no finding of ARDS, were not receiving immunosuppressive therapy or had no documented HIV positivity were included in the study. It was stipulated that RT-PCR tests for both disease groups be performed for patients diagnosed after March 2020.

**Exclusion criteria.** Patients were excluded if they presented with no lesions on thorax CT or smaller than 1 mL, positive results in both PCR test group, atypical findings (cavity, pleural effusion, tree-in-bud pattern), or severe respiratory motion artifacts were exclusion criteria (Fig 1).

As a result, 74 AP and 60 COVID-19 patients finally constituted the study population.

**CT acquisition.** All thorax CT studies were performed with a 128-detector system (GE Revolution, General Electric, Milwaukee, WI), from the first rib to the adrenal glands, nonenhanced, using the following parameters: 100 kV, 110 mAs, body filter, 1.25 mm slice thickness, 512x512 reconstruction matrix, spiral pitch factor 1.375:1, BonePlus convolution kernel, adaptive statistical iterative reconstruction 70%.

**Radiomics parameter calculation.** Lesions were manually sampled by two radiologists (MG, 15 years of experience and BARM, last year of radiology residency) separately using Olea Sphere 3.0 SP 21 software (Olea Medicals, La Ciotat, France) and the same rules: 1. One individual lesion is segmented from one end to the other at a time; 2. If a septal, vascular or bronchial structure is totally encased by the lesion, it is included in the segmentation; and 3. If the septal, vascular or bronchial structure remains on the surface of the lesion, it is not sampled.

Size, Shape, First-order and Second-order (Gray Level Run Length Matrix (GLRLM), Gray Level Size Zone Matrix (GLSZM), Gray Level Dependence Matrix (GLDM), Gray Level Co-occurrence Matrix (GLCM) and Neighboring Gray Tone Difference Matrix (NGTDM)) radiomic parameters were calculated using the volume of interest (VOI) of the manual segmentations as previously described [21].

The slices were resampled to a voxel size of 0.8x0.8x1.25 $mm^3$ by using bicubic interpolation for intensity. The number of bins for histogram preparation and gray-level discretization was set to 64.

Voxel densities were not used as Hounsfield units (HU) but were normalized according to Eq 1:

$$f(x) = \frac{s(x - \mu_x)}{\sigma_x} \tag{1}$$

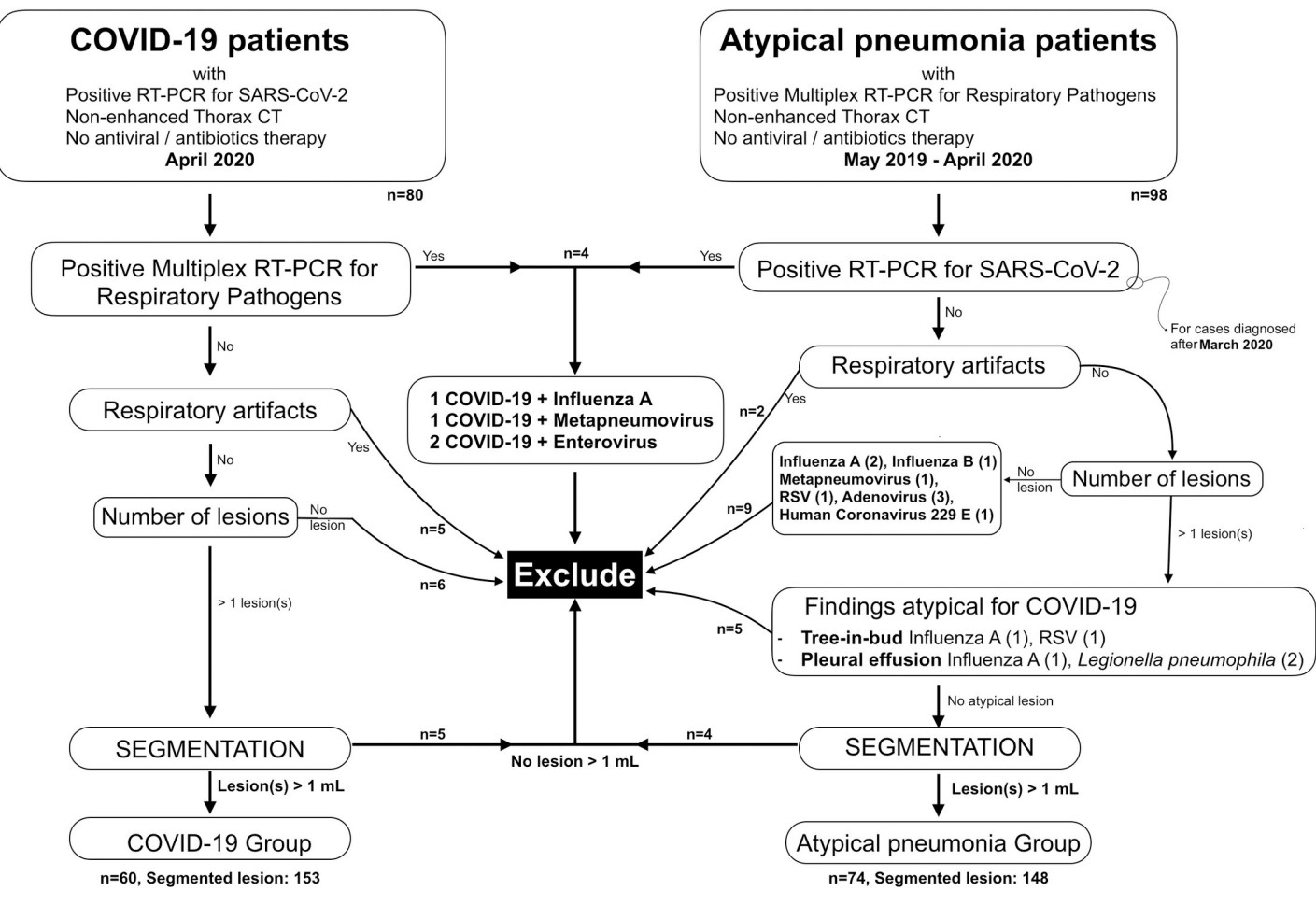

**Fig 1. Flowchart of the study.**

where f(x) is the normalized voxel density, s is a scaling factor (set to 1), x is the original density, $\mu_x$ is the mean density calculated from *all voxels of the slice in and outside of the segmentation* and $\sigma_x$ is the standard deviation.

We mainly worked with negative HU values and avoided adding a fixed positive integer (voxel array shift) to the measured HU values for the calculation of the total energy parameter, thus avoiding the volume confounding effect. Neighbor distance was set to 1 mm and examined 0˚, 45˚, 90˚ and 135˚ from the center voxel isotropically (13 directions). For the dependence matrices, neighboring voxels were considered dependent on the center voxel if both were equal in gray level.

**Statistical analysis.** Lesions were classified as ground-glass opacities (GGOs), crazy paving signs (CPs) or consolidations by the two radiologists (MG and BARM) separately. If there was inconsistency in the classification, a joint decision was made. All pneumonic lesions present in patients were classified, and all classified lesions were used for segmentation, feature extraction and parameter calculation and statistical evaluation.

GGOs (n = 150), CPs (n = 52) and consolidations (n = 99) groups were evaluated separately to assess the differences in the parameters as they related to COVID-19 and AP. Since there were multiple outliers, especially in the parameters correlated strongly with volume, a nonnormal distribution was very often observed in the group comparisons. A logarithmic

transformation was performed, and the T-test was used if the transformed data were normally distributed. Otherwise, the Kruskal-Wallis and post hoc Mann-Whitney U tests were used. The First-order parameters were also evaluated for their ability to distinguish between different lesion types.

The assumption of linearity for the parameters in the model was evaluated with the Box-Tidwell procedure. The adequacy of the model parameters in predicting the categorical outcomes was evaluated with the Hosmer-Lemeshow goodness of fit test, and p>0.05 was evaluated as a good fit. Multicollinearity of the parameters in the models (such as between Range and Interquartile range or Flatness and Spherical Disproportion) was evaluated by the variance inflation factor (VIF) of the linear regression test.

Model validation was performed using leave one out cross-validation (LOOCV). For this purpose, all lesions belonging to the same patient were turned into a separate block and formed the test group, and the remaining lesions comprised the training group. Thus, validation was conducted 134 times for the GGO and consolidation group, 86 times for the positive skewness group (PSG) and 68 times for the negative skewness group (NSG).

Statistical analyses were performed using IBM SPSS v23 (IBM Corp, Armonk, NY), Med-Calc v14.8.1 (MedCalc Software bvba, Ostend, Belgium), R v4.0.2 (R foundation, Vienna, Austria) and the XLStat statistical and data analysis add-on 2020.3.1 (Addinsoft, NY, USA) for Microsoft Excel 16.0.13029. Power analysis was conducted using G*Power 3.1 (Faul, Erdfelder, Lang, & Buchner, 2007).

## Results

### Study population

There were 60 COVID-19 (39 male and 21 female; 66.6% and 33.3%, respectively) and 74 AP patients (46 male and 28 female; 62.2% and 37.8%, respectively) included in the study. The mean age was 48.25±13.67 (22–83 years) in the COVID-19 group and 48.01 ±20.38 (16–96 years) in the AP group. No significant difference was found in terms of sex (p = 0.573, chi-square test) or age (p = 0.921, T-test) between the groups.

The AP group consisted of patents with influenza A (n = 17), influenza B (n = 9), adenovirus (n = 10), human coronavirus (HCoV-229E n = 1; HCoV-NL63 n = 2; HCoV-HKU-1 n = 1), metapneumovirus (n = 11), respiratory syncytial virus (RSV) (n = 6), *Mycoplasma pneumoniae* (n = 15), and *Legionella pneumophila* (n = 2).

### Pulmonary lesions

There were 153 (50.8%) COVID-19 (1–6 lesions/patient; mean 2.55) and 148 (49.2%) AP lesions (1–4 lesions/patient; mean 2.00) obtained from the study population (Table 1). The limited number of segmented lesions used in the study was based on two reasons: (1) Lesions less than 1 mL in volume were not used in the study, and (2) Lesions that merged with each other were processed as a single lesion. No significant difference was found in the number of GGO lesions between the COVID-19 and AP groups (p = 0.721, post hoc z-score test), although there was a significant difference in the number of CP (p = 0.000) and consolidation (p = 0.003) lesions.

The time from documented first high fever or hospitalization to CT scan was slightly higher in the AP group (7.7 [1–14] days) than in the COVID-19 group (4.4 [1–10] days).

Two authors (MG and BARM) segmented lesions separately using the same rules. Mean volume (26.855±34.445 mL and 28.211±37.092 mL, respectively) and mean density (0.590 ±0.162 and 0.579±173) of the segmentations were compared using Bland Altman analysis and paired samples T-test and no significant difference was found for calculated mean volume (p = 0.589) and density (p = 0.154).

**Table 1. Distribution of lesions according to patient groups, lesion types and lobes.**

| Location | COVID (n = 153) | | | Atypical Pneumonia (n = 148) | | | TOTAL |
|---|---|---|---|---|---|---|---|
| | GGO | CP | Consolidation | GGO | CP | Consolidation | |
| RUL | 12 | 2 | 7 | 11 | 2 | 12 | 46 |
| RML | 6 | 2 | 0 | 7 | 0 | 3 | 18 |
| RLL | 28 | 17 | 15 | 23 | 4 | 16 | 103 |
| LUL | 14 | 9 | 7 | 10 | 4 | 11 | 55 |
| LLL | 16 | 8 | 10 | 23 | 4 | 18 | 79 |
| TOTAL | 76 | 38 | 39 | 74 | 14 | 60 | 301 |

GGO: Ground glass opacities, CP: Crazy paving, RUL: Right upper lobe, RML: Right middle lobe, RLL: Right lower lobe, LUL: Left upper lobe (including lingular lobe), LLL: Left lower lobe.

## Radiomics parameters

Size, Shape, Location, First-order (Tables 2 and 3) and Second-order (Tables 4 and 5) texture parameters were evaluated.

## Shapes, sizes and locations of the lesions

The volumes of the lesions were spread over a wide range (1.235–165.504 mL in the COVID-19 group and 1.022–176.192 mL in the AP group) with nonnormal distributions and wide variability (mean 23.565 ± 32.181 mL in the COVID-19 group and 30.756 ± 36.696 mL in the AP group). However, there was no significant difference in the lesion volume between the COVID-19 and AP groups (p = 0.084, Kruskal-Wallis test).

The Shape parameters Sphericity, Compactness, Spherical Disproportion, Elongation and Flatness demonstrated a tendency of the COVID-19 lesions to be more rounded (Table 3). Receiver operating characteristic (ROC) analysis showed that the Shape and Size parameters individually had poor sensitivity, specificity and AUC values in discriminating lesions (Table 3).

In the COVID-19 group 105 lesions (68.6%) were located peripherally, 27 lesions (17.7%) were located centrally and 21 lesions (13.7%) were located diffusely; in the AP group, these numbers were 60 (40.5%), 27 (18.2%) and 61 (41.2%), respectively. The number of lesions of each location type were significantly different between the groups (p = 0.000, chi-square test). In our study group, the cause of the pneumonic lesions was correctly predicted for 64.5% of the lesions (sensitivity 60.1% and specificity 68.6%) by using their location data (peripheral or nonperipheral) only.

## First-order texture parameters

The First-order texture parameters' discriminability of the whole group of COVID-19 and AP lesions was poor (Table 2); however, these parameters were found to be effective in categorizing pneumonic lesions alone (Table 3). While consolidations yielded negative skewness unless they had extensive ground-glass halo areas (Fig 2), GGO lesions with a volume greater than 3.0 mL had positive skewness values. GGO lesions smaller than this volume showed negative skewness, and skewness maps showed that voxels corresponding to enlarged septal or vascular structures led to a right shift (Fig 3). The mean skewness was found to be significantly different among GGO, CP and consolidation lesions.

Several parameters were found to be significantly different for GGO and consolidation lesions between COVID-19 and AP (Table 3), and Range was the only parameter that could

**Table 2. ROC analysis results of size, shape and first order texture parameters to discriminate COVID-19 related lesions from atypical pneumonia.**

| Parameter | COVID-19 (mean ± SD) | Atypicala* (mean ± SD) | Cut-off | Sensitivity (%) | Specificity (%) | AUC | 95% CI |
|---|---|---|---|---|---|---|---|
| **Size and Shape Parameters** | | | | | | | |
| Sphericity | 0,446±0,110 | 0,390±0,084 | >0,405 | 60,1 | 64,3 | 0,643 | 0,586–0,698 |
| Compactness 1 | 0,016±0,006 | 0,013±0,006 | >0,013 | 60,8 | 63,6 | 0,637 | 0,579–0,692 |
| Compactness 2 | 0,104±0,079 | 0,068±0,045 | >0,109 | 36,6 | 86,7 | 0,633 | 0,575–0,688 |
| Elongation | 0,685±0,150 | 0,648±0,144 | >0,754 | 37,9 | 79,0 | 0,579 | 0,520–0,636 |
| Flatness | 0,477±0,154 | 0,412±0,130 | >0,493 | 46,4 | 74,1 | 0,614 | 0,556–0,670 |
| Surface area (cm$^2$) | 9,23±9,49 | 12,45±11,22 | ≤8,35 | 67,3 | 51,8 | 0,600 | 0,541–0,656 |
| Area/Volume ratio | 0,508±0,135 | 0,532±0,166 | ≤0,638 | 85,6 | 21,7 | 0,529 | 0,471–0,587 |
| Spherical disproportion | 2,385±0,598 | 2,690±0,594 | ≤2,572 | 68,0 | 57,3 | 0,645 | 0,588–0,700 |
| Major axis (mm) | 53,08±23,52 | 50,42±26,47 | ≤39,68 | 41,8 | 74,8 | 0,596 | 0,538–0,652 |
| Minor axis (mm) | 35,76±14,61 | 32,53±14,75 | ≤36,42 | 69,9 | 45,5 | 0,571 | 0,513–0,628 |
| Least axis (mm) | 22,39±10,27 | 21,63±9,00 | ≤22,66 | 67,3 | 45,5 | 0,512 | 0,454–0,570 |
| **First Order Texture Parameters** | | | | | | | |
| Skewness | 0,152±0,489 | 0,114±0,672 | >-0,331 | 85,0 | 28,0 | 0,519 | 0,461–0,577 |
| Kurtosis | 3,356±0,961 | 3,460±1,370 | ≤3,107 | 53,0 | 55,0 | 0,504 | 0,446–0,563 |
| Energy | 9019±11856 | 17030±22052 | ≤3408 | 47,0 | 70,6 | 0,615 | 0,557–0,671 |
| Total Energy | 8222±10776 | 16258±22125 | ≤7846 | 69,3 | 48,3 | 0,608 | 0,550–0,664 |
| Entropy | 5,153±0,240 | 5,136±0,258 | >5,135 | 59,5 | 53,9 | 0,527 | 0,468–0,585 |
| Minimum** | 0,073±0,077 | 0,058±0,096 | >-0,028 | 92,2 | 20,3 | 0,526 | 0,467–0,584 |
| 10th percentile** | 0,354±0,116 | 0,380±0,142 | ≤0,496 | 86,9 | 25,2 | 0,554 | 0,496–0,612 |
| 90th percentile** | 0,768±0,161 | 0,857±0,207 | ≤0,889 | 74,5 | 49,7 | 0,637 | 0,579–0,692 |
| Maximum** | 1,221±0,192 | 1,344±0,229 | ≤1,342 | 81,7 | 50,3 | 0,672 | 0,615–0,725 |
| Mean** | 0,561±0,139 | 0,622±0,179 | ≤0,668 | 75,8 | 42,0 | 0,603 | 0,544–0,659 |
| Median** | 0,561±0,149 | 0,625±0,202 | ≤0,742 | 86,9 | 31,5 | 0,590 | 0,531–0,646 |
| Range** | 1,149±0,199 | 1,284±0,234 | ≤1,321 | 86,9 | 45,5 | 0,686 | 0,630–0,739 |
| Interquartile Range** | 0,220±0,057 | 0,257±0,085 | ≤0,296 | 92,1 | 26,6 | 0,622 | 0,564–0,677 |
| Standard deviation | 0,161±0,035 | 0,187±0,045 | ≤0,161 | 50,3 | 74,8 | 0,664 | 0,607–0,717 |
| Mean absolute deviation | 0,129±0,030 | 0,151±0,041 | ≤0,165 | 89,5 | 33,6 | 0,649 | 0,591–0,703 |
| Robust mean deviation | 0,091±0,023 | 0,107±0,034 | ≤0,121 | 92,2 | 28,7 | 0,630 | 0,572–0,685 |
| Root mean squared | 0,585±0,138 | 0,651±0,179 | ≤0,657 | 71,9 | 47,6 | 0,612 | 0,554–0,668 |
| Variance | 0,027±0,011 | 0,037±0,017 | ≤0,027 | 52,3 | 72,7 | 0,666 | 0,609–0,720 |
| Uniformity | 0,035±0,008 | 0,033±0,006 | ≤0,034 | 66,7 | 48,3 | 0,552 | 0,493–0,609 |

*Atypical: Atypical pneumonia group.

** Standardized data. Figures were not given in Hounsfield unit.

discriminate all lesion types in both disease groups and had the best AUC in ROC analysis, although its specificity was 45.5%.

## Second-order texture parameters

The Second-order texture parameters' discriminability of the COVID-19 and AP lesions was also poor (Table 4). Only the parameters Large Area Low Gray Level Emphasis and GLCM-Correlation had AUCs greater than 0.600, although their sensitivity in differentiating COVID-19 lesions was merely 50%.

None of the Second-order parameters showed a significant difference for the CP lesions between the COVID-19 and AP groups. However, there were parameters significantly different in GGO and consolidation (Table 5).

**Table 3. Level of statistical significance (p values) of the first order texture parameters between the pneumonic lesion types of whole study population and between the same type lesions of COVID-19 and atypical pneumonia groups.**

| Parameter | Pneumonic Lesion Comparison | | | Comparison of COVID-19 and Atypical Pneumonia | | |
|---|---|---|---|---|---|---|
| | GGO and Cons. | GGO and CP | CP and Cons. | GGO | Consolidation | CP |
| Skewness | 0,000 [1] | 0,000 [1] | 0,000 [1] | 0,007 [2] | 0,288 [2] | 0,396 [2] |
| Kurtosis | 0,000 [2] | 0,000 [2] | 0,643 [2] | 0,940 [2] | 0,642 [2] | 0,567 [2] |
| Energy | 0,000 [2] | 0,045 [2] | 0,000 [2] | 0,265 [2] | 0,000 [2] | 0,526 [2] |
| Total Energy | 0,000 [2] | 0,027 [2] | 0,003 [2] | 0,353 [2] | 0,001 [2] | 0,821 [2] |
| Entropy | 0,015 [1] | 0,000 [1] | 0,260 [1] | 0,175 [2] | 0,051 [2] | 0,203 [2] |
| Minimum | 0,024 [2] | 0,070 [2] | 0,548 [2] | 0,440 [2] | 0,678 [2] | 0,158 [2] |
| 10th percentile | 0,000 [2] | 0,209 [2] | 0,000 [2] | 0,772 [2] | 0,241 [2] | 0,880 [2] |
| 90th percentile | 0,000 [1] | 0,000 [1] | 0,000 [1] | 0,014 [2] | 0,000 [2] | 0,795 [2] |
| Maximum | 0,000 [1] | 0,000 [1] | 0,984 [1] | 0,000 [3] | 0,000 [3] | 0,086 [3] |
| Mean | 0,000 [1] | 0,000 [1] | 0,000 [1] | 0,373 [3] | 0,003 [3] | 0,781 [3] |
| Median | 0,000 [1] | 0,000 [1] | 0,000 [1] | 0,750 [3] | 0,002 [3] | 0,781 [3] |
| Range | 0,000 [1] | 0,000 [1] | 0,962 [1] | 0,000 [3] | 0,000 [3] | 0,042 [3] |
| Interquartile Range | 0,000 [2] | 0,000 [2] | 0,948 [2] | 0,002 [3] | 0,000 [3] | 0,625 [3] |
| Standard deviation | 0,000 [1] | 0,000 [1] | 0,808 [1] | 0,000 [3] | 0,000 [3] | 0,423 [3] |
| Mean absolute deviation | 0,000 [2] | 0,000 [2] | 0,889 [2] | 0,001 [3] | 0,000 [3] | 0,503 [3] |
| Robust mean deviation | 0,000 [2] | 0,000 [2] | 0,873 [2] | 0,001 [3] | 0,000 [3] | 0,597 [3] |
| Root mean squared | 0,000 [1] | 0,000 [1] | 0,000 [1] | 0,203 [3] | 0,001 [3] | 0,785 [3] |
| Variance | 0,000 [2] | 0,000 [2] | 0,738 [2] | 0,000 [3] | 0,000 [3] | 0,474 [3] |
| Uniformity | 0,166 [1] | 0,000 [1] | 0,030 [1] | 0,357 [2] | 0,027 [2] | 0,203 [2] |

1. Results of ANOVA.

2. Results of Mann Whitney test.

3. Results of T-test.

GGO: Ground glass opacities, Cons: Consolidation, CP: Crazy paving.

## Models for lesion estimation

Logistic regression probabilistic models were developed since no individual parameter showed good discriminability. A total of 18 First and Second-order parameters discriminated both GGO and consolidation (Tables 3 and 5) lesions, and no parameters other than Range could discriminate CP lesions between COVID-19 and AP. Thus, models focusing on GGOs and consolidations were built. These 18 parameters were merged with the Shape and Size parameters with AUCs greater than 0.600 (Table 2) as well as the Location parameter; thus, a total of 23 parameters were used to generate the models. The parameters were logarithmically transformed to prevent the model from being influenced by outliers and the skewness values of the parameters.

All possible three- and four-parameter combinations were studied for candidate models. No more than four parameters were used to build the models to prevent overfitting. The Bayesian information criterion (BIC) estimator was used to select the best among the candidate models

The group consisting of all consolidation and GGO lesions was the largest group and included 134 patients and 249 lesions; the best three- and four-parameter models that predicted both types of lesion showed modest sensitivity and specificity for both the training and test sets (Models 1 and 2, Table 6). Further evaluation of the subgroups revealed that the accuracy for COVID-19 consolidations were 46.1% with Model-1 and 56.4% with Model-2.

**Table 4. ROC analysis results of second order texture parameters in discrimination of COVID-19 from atypical pneumonia.**

| Parameter | COVID-19 (mean ± SD) | Atypical* (mean ± SD) | Cut-off | Sensitivity (%) | Specificity (%) | AUC | 95% CI |
|---|---|---|---|---|---|---|---|
| **Gray Level Run Length Matrix** | | | | | | | |
| Short run emphasis | 0,96801±0,00662 | 0,96654±0,00617 | >0,96590 | 66,0 | 51,8 | 0,578 | 0,520–0,635 |
| Long run emphasis | 1,14102±0,03274 | 1,15699±0,07290 | ≤1,14780 | 66,0 | 51,8 | 0,580 | 0,522–0,637 |
| Gray level non-uniformity | 1190,53±1397,49 | 881,20±1334,74 | ≤1086,81 | 83,0 | 37,8 | 0,592 | 0,534–0,648 |
| GLNUN | 0,03302±0,00645 | 0,03410±0,00741 | ≤0,03300 | 60,1 | 53,9 | 0,545 | 0,487–0,603 |
| Run length non-uniformity | 23506,4±34546,5 | 30076,7±34116,6 | ≤26708,3 | 79,1 | 37,8 | 0,588 | 0,529–0,644 |
| RLNUN | 0,91802±0,01652 | 0,91258±0,02332 | >0,91380 | 66,0 | 51,1 | 0,578 | 0,519–0,635 |
| Run Percentage | 0,95683±0,00918 | 0,95325±0,01550 | >0,95460 | 66,7 | 51,8 | 0,579 | 0,520–0,636 |
| Gray Level Variance | 84,942±26,894 | 93,279±39,923 | ≤125,813 | 93,5 | 17,5 | 0,527 | 0,469–0,585 |
| Run variance | 0,04822±0,01149 | 0,05508±0,03371 | ≤0,05080 | 66,0 | 52,5 | 0,581 | 0,523–0,638 |
| Run entropy | 5,4190±0,2048 | 5,4232±0,2248 | >5,4034 | 62,1 | 51,8 | 0,508 | 0,450–0,566 |
| Short run LGLE | 0,00409±0,00345 | 0,00441±0,00418 | ≤0,00120 | 2,61 | 90,9 | 0,510 | 0,451–0,568 |
| Short run HGLE | 864,31±343,07 | 918,48±383,84 | ≤601,51 | 27,45 | 84,7 | 0,536 | 0,478–0,594 |
| Long run LGLE | 0,00476±0,00438 | 0,00520±000499 | ≤0,00460 | 68,6 | 39,9 | 0,521 | 0,462–0,579 |
| Long run HGLE | 1032,74±610,32 | 1018,60±406,67 | ≤1396,87 | 83,0 | 27,9 | 0,540 | 0,482–0,598 |
| **Gray Level Size Zone Matrix** | | | | | | | |
| Small area emphasis | 0,73030±0,02891 | 0,73069±0,02926 | >0,72590 | 57,5 | 53,2 | 0,523 | 0,463–0,580 |
| Large area emphasis | 11,203±15,588 | 21,896±34,501 | ≤8,336 | 64,1 | 55,2 | 0,598 | 0,540–0,655 |
| Gray level non-uniformity | 416,24±580,31 | 524,29±580,31 | ≤490,43 | 80,4 | 37,8 | 0,585 | 0,526–0,641 |
| GLNUN | 0,02914±0,00431 | 0,02905±0,00465 | ≤0,02930 | 62,1 | 50,4 | 0,501 | 0,442–0,559 |
| Size zone non-uniformity | 6710,49±9451,04 | 8376,19±9173,09 | ≤5542,69 | 68,6 | 46,9 | 0,591 | 0,533–0,648 |
| SZNUN | 0,49444±0,39484 | 0,49509±0,04116 | >0,48730 | 57,5 | 53,2 | 0,521 | 0,462–0,579 |
| Zone Percentage | 0,55550±0,07183 | 0,54358±0,07541 | >0,54180 | 60,8 | 52,5 | 0,559 | 0,501–0,617 |
| Gray Level Variance | 100,27±24,43 | 106,38±34,39 | ≤135,39 | 92,8 | 18,9 | 0,517 | 0,458–0,575 |
| Zone variance | 7,4490±12,2950 | 15,6470±24,2230 | ≤5,4620 | 68,6 | 50,4 | 0,599 | 0,541–0,656 |
| Zone entropy | 6,8400±0,1761 | 6,8651±0,1637 | ≤6,7820 | 39,2 | 71,3 | 0,533 | 0,474–0591 |
| Small area LGLE (x10⁻³) | 4,0882±3,9304 | 3,9978±3,0079 | >2,0311 | 77,1 | 33,6 | 0,527 | 0,469–0,586 |
| Small area HGLE | 641,65±230,14 | 676,65±234,09 | ≤456,78 | 26,1 | 87,4 | 0,546 | 0,487–0,603 |
| Large area LGLE | 0,04461±0,09193 | 0,08423±0,18103 | ≤0,01420 | 48,4 | 69,9 | 0,611 | 0,553–0,667 |
| Large area HGLE | 3,8300±0,3120 | 4,0520±0,7410 | ≤4,1680 | 91,5 | 27,3 | 0,546 | 0,487–0,603 |
| **Gray Level Dependence Matrix** | | | | | | | |
| Small dependence emphasis | 0,48389±0,06201 | 0,47551±0,06548 | >0,5037 | 41,8 | 72,7 | 0,561 | 0,503–0,619 |
| Large dependence emphasis | 5,9591±1,4232 | 6,5414±2,1320 | ≤6,3424 | 71,2 | 49,0 | 0,587 | 0,529–0,644 |
| Gray level non-uniformity | 935,11±1420,84 | 1283,38±1515,39 | ≤1138,71 | 82,4 | 38,5 | 0,594 | 0,536–0,651 |
| Dependence non-uniformity | 7298,03±10526,77 | 9013,57±10131,83 | ≤8895,23 | 79,7 | 39,7 | 0,584 | 0,526–0,641 |
| DNUN | 0,28855±0,04086 | 0,27925±0,04861 | >0,27260 | 65,4 | 49,0 | 0,573 | 0,514–0,630 |
| Gray level variance | 92,904±40,013 | 83,956±27,032 | ≤124,49 | 92,8 | 18,1 | 0,527 | 0,469–0,585 |
| Dependence Variance | 1,3980±0,3689 | 1,6304±0,7358 | ≤1,4604 | 64,7 | 56,6 | 0,598 | 0,539–0,654 |
| Dependence entropy | 7,0767±0,1866 | 7,1179±0,2015 | ≤6,9314 | 24,2 | 88,1 | 0,538 | 0,479–0,596 |
| Small dependence LGLE | 0,002497±0,00181 | 0,002499±0,00238 | >0,00070 | 96,7 | 10,5 | 0,525 | 0,466–0,583 |
| Small dependence HGLE | 432,60±176,85 | 440,46±157,71 | ≤299,44 | 30,1 | 85,3 | 0,521 | 0,462–0,579 |
| Large dependence LGLE | 0,02631±0,04705 | 0,03021±0,04296 | ≤0,0227 | 77,8 | 37,1 | 0,575 | 0,516–0,632 |
| Large dependence HGLE | 5302,01±2507,58 | 8395,02±16774,12 | ≤6592,00 | 81,7 | 32,9 | 0,533 | 0,474–0,590 |
| **Gray Level Co-occurrence Matrix** | | | | | | | |
| Contrast | 79,343±24,509 | 78,981±27,491 | >77,212 | 52,9 | 60,1 | 0,523 | 0,464–0,581 |
| Correlation | 0,47346±0,09845 | 0,50980±0,11269 | ≤0,46600 | 50,3 | 69,9 | 0,609 | 0,551–0,665 |
| Maximum probability (x10⁻³) | 5,1548±2,9506 | 5,5979±6,0588 | ≤3,9018 | 50,3 | 62,9 | 0,567 | 0,508–0,624 |

*(Continued)*

**Table 4.** (Continued)

| Parameter | COVID-19 (mean ± SD) | Atypical* (mean ± SD) | Cut-off | Sensitivity (%) | Specificity (%) | AUC | 95% CI |
|---|---|---|---|---|---|---|---|
| Autocorrelation | 906,23±370,89 | 954,55±428,98 | ≤632,95 | 28,1 | 81,1 | 0,527 | 0,468–0,585 |
| Cluster prominence (x10³) | 17,822±25,716 | 25,484±24,242 | ≤33,817 | 94,1 | 25,2 | 0,567 | 0,509–0,625 |
| Cluster shade | 211,56±1730,93 | 89,101±3981,97 | >-3072,76 | 99,4 | 14,0 | 0,502 | 0,443–0,560 |
| Cluster tendency | 230,030±82,402 | 267,822±134,992 | ≤319,011 | 86,9 | 28,7 | 0,552 | 0,494–0,610 |
| Difference average | 6,7681±1,1229 | 6,6504±1,1580 | >6,7933 | 51,0 | 63,6 | 0,550 | 0,492–0,608 |
| Difference entropy | 4,1817±0,2246 | 4,1648±0,2158 | >4,1974 | 52,3 | 60,8 | 0,542 | 0,483–0,600 |
| Difference variance | 31,743±9,224 | 32,813±10,519 | ≤22,196 | 37,9 | 72,7 | 0,519 | 0,461–0,577 |
| IDMN | 0,98251±0,00518 | 0,98265±0,00563 | ≤0,97880 | 52,3 | 61,5 | 0,526 | 0,467–0,584 |
| IDN | 0,911045±0,01294 | 0,912721±0,01319 | ≤0,9098 | 48,4 | 67,8 | 0,557 | 0,499–0,615 |
| Inverse variance | 0,152834±0,02489 | 0,160731±0,02848 | ≤0,15430 | 60,1 | 57,3 | 0,593 | 0,535–0,650 |
| **Neighboring Gray Tone Difference Matrix** | | | | | | | |
| Coarseness (x10⁻³) | 1,34299±1,69595 | 0,93792±1,14415 | >1,04190 | 34,6 | 76,9 | 0,562 | 0,503–0,619 |
| Contrast | 0,2137±0,0911 | 0,2174±0,1139 | >0,1844 | 57,5 | 51,1 | 0,513 | 0,454–0,571 |
| Busyness | 2,0291±3,8506 | 2,2528±3,0269 | ≤2,5093 | 85,0 | 30,0 | 0,581 | 0,522–0,637 |
| Complexity | 7544,73±1387,75 | 7451,53±1219,89 | >8067,22 | 41,2 | 75,5 | 0,542 | 0,483–0,600 |
| Strength | 1,6840±1,8716 | 1,5113±1,8949 | >1,1493 | 44,4 | 65,0 | 0,542 | 0,483–0,599 |

* Atypical: Atypical pneumonia group, GLNUN: Gray Level Non-uniformity Normalized, RLNUN: Run Length Non-uniformity Normalized, LGLE: Low Gray Level Emphasis, HGLE: High Gray Level Emphasis, SZNUN: Size Zone Non-uniformity Normalized, DNUN: Dependency Non-uniformity Normalized, IDMN: Inverse Difference Moment Normalized, IDN: Inverse Difference Normalized.

Similarly, both models had low accuracy for AP-related GGO lesions (60.8% and 54.1%, respectively). The high accuracy in predicting AP-related consolidation lesions (90% and 83.3%) and COVID-19-related GGO lesions (82.9% and 81.5%), which both constituted 54.6% of the lesions, appeared the models more successful than they actually were.

Since low accuracy affected the consolidation and GGO subgroups in the single model approaches, we decided to study them separately. In our study, there were only a few pure consolidations (7 COVID-19, 6 influenza, 6 adenovirus and 1 *Legionella pneumophila*-associated lesions), and almost all lesions were including both ground-glass and consolidation areas which posed a classification problem. We concluded that separating the lesions according to their skewness values would eliminate the need for a subjective decision-maker; thus, the lesions were grouped into the PSG (n = 149) and NSG (n = 100) according to their skewness values.

The PSG included 78 lesions (52.3%) from 49 patients with AP and 71 (47.7%) lesions from 37 COVID-19 patients. While 142 of these lesions were GGOs, 7 were consolidations with wide ground-glass halo. The best models for PSG lesion prediction always included the parameters GLCM-Contrast and Range. The BIC analysis showed that the best 3-parameter model was obtained by adding Sphericity (Model-3, Table 6). Higher values of the parameters GLCM-Contrast and Range increased the likelihood of the lesion being identified as an AP lesion; in contrast, a higher value for the Sphericity parameter increased the likelihood of the lesion being identified as a COVID-19 lesion. There were 20 COVID-19 and 11 AP lesions with the Sphericity value was greater than 0.500, and Model-3 correctly predicted 17 (85.0%) COVID-19 and 8 (72.7%) AP lesions, showing that the model had no tendency to classify the most rounded lesions as COVID-19. In the cross-validation study, Model-3 had the best sensitivity, specificity and accuracy for PSG lesions (Table 6).

When the parameters evaluating the shape of the lesion were not used during model creation, the best model included the Lesion location parameter (Model-4, Table 6). Such models

**Table 5. Level of statistical significance (p values) of the second order texture parameters between the same type lesions of COVID-19 and atypical pneumonia groups.**

| Parameter | GGO | CP | Consolidation |
|---|---|---|---|
| **Gray Level Run Length Matrix** | | | |
| Short run emphasis | 0,024 [1] | 1,000 [1] | 1,000 [1] |
| Long run emphasis | 0,010 [1] | 1,000 [1] | 0,146 [1] |
| Gray level non-uniformity | 0,016 [1] | 1,000 [1] | 0,088 [1] |
| GLNUN | 0,250 [1] | 0,276 [2] | 0,029 [2] |
| Run length non-uniformity | 0,120 [1] | 1,000 [1] | 0,600 [1] |
| RLNUN | 0,012 [1] | 1,000 [1] | 0,169 [1] |
| Run Percentage | 0,011 [1] | 1,000 [1] | 0,151 [1] |
| Gray Level Variance | 0,210 [2] | 0,073 [2] | 0,872 [2] |
| Run variance | 0,009 [1] | 1,000 [1] | 0,136 [1] |
| Run entropy | 0,088 [2] | 0,100 [2] | 0,241 [2] |
| Short run LGLE | 0,895 [1] | 0,372 [1] | 0,753 [1] |
| Short run HGLE | 0,752 [2] | 0,941 [2] | 0,857 [2] |
| Long run LGLE | 0,766 [1] | 0,304 [1] | 0,596 [1] |
| Long run HGLE | 0,960 [1] | 0,941 [2] | 0,529 [2] |
| **Gray Level Size Zone Matrix** | | | |
| Small area emphasis | 0,036 [1] | 0,430 [2] | 0,025 [2] |
| Large area emphasis | 0,137 [1] | 0,343 [1] | 0,000 [1] |
| Gray level non-uniformity | 0,473 [1] | 0,480 [1] | 0,006 [1] |
| GLNUN | 0,260 [2] | 0,080 [2] | 0,798 [2] |
| Size zone non-uniformity | 0,276 [1] | 0,585 [1] | 0,533 [1] |
| SZNUN | 0,011 [2] | 0,406 [2] | 0,022 [2] |
| Zone Percentage | 0,011 [2] | 0,484 [2] | 0,000 [1] |
| Gray Level Variance | 0,041 [1] | 0,089 [1] | 0,636 [1] |
| Zone variance | 0,228 [1] | 0,418 [1] | 0,000 [1] |
| Zone entropy | 0,583 [2] | 0,408 [2] | 0,018 [2] |
| Small area LGLE | 1,000 [1] | 1,000 [1] | 1,000 [1] |
| Small area HGLE | 0,329 [2] | 0,206 [2] | 0,429 [2] |
| Large area LGLE | 1,000 [1] | 1,000 [1] | 0,000 [1] |
| Large area HGLE | 0,054 [1] | 1,000 [1] | 0,572 [1] |
| **Gray Level Dependence Matrix** | | | |
| Small dependence emphasis | 0,019 [2] | 0,513 [2] | 0,002 [1] |
| Large dependence emphasis | 0,087 [2] | 0,616 [2] | 0,001 [1] |
| Gray level non-uniformity | 1,000 [1] | 1,000 [1] | 0,061 [1] |
| Dependence non-uniformity | 1,000 [1] | 1,000 [1] | 0,402 [1] |
| DNUN | 0,014 [2] | 0,392 [2] | 0,001 [2] |
| Gray level variance | 0,293 [1] | 1,000 [1] | 1,000 [1] |
| Dependence Variance | 0,220 [2] | 0,603 [2] | 1,000 [1] |
| Dependence entropy | 0,911 [2] | 0,265 [2] | 0,395 [1] |
| Small dependence LGLE | 1,000 [1] | 0,691 [2] | 0,831 [2] |
| Small dependence HGLE | 1,000 [1] | 0,242 [2] | 0,021 [2] |
| Large dependence LGLE | 0,017 [2] | 1,000 [1] | 0,143 [1] |
| Large dependence HGLE | 0,899 [1] | 0,622 [2] | 1,000 [1] |
| **Gray Level Co-occurrence Matrix** | | | |
| Contrast | 0,004 [1] | 0,209 [2] | 0,008 [2] |
| Correlation | 1,000 [1] | 0,430 [2] | 0,001 [2] |

*(Continued)*

**Table 5.** (Continued)

| Parameter | GGO | CP | Consolidation |
|---|---|---|---|
| Maximum probability | 1,000 [1] | 0,565 [2] | 0,245 [1] |
| Autocorrelation | 0,620 [2] | 0.210 [2] | 0,920 [2] |
| Cluster prominence | 0,026 [1] | 1,000 [1] | 1,000 [1] |
| Cluster shade | 0,170 [1] | 0,625 [2] | 1,000 [1] |
| Cluster tendency | 0,171 [1] | 0,138 [2] | 1,000 [1] |
| Difference average | 0,001 [2] | 0,258 [2] | 0,000 [2] |
| Difference entropy | 0,000 [2] | 0,239 [2] | 0,007 [1] |
| Difference variance | 0,000 [1] | 0,078 [2] | 0,358 [1] |
| IDMN | 0,004 [1] | 0,152 [2] | 0,003 [2] |
| IDN | 0,003 [2] | 0,248 [2] | 0,000 [2] |
| Inverse variance | 0,071 [2] | 0,441 [2] | 0,000 [1] |
| **Neighboring Gray Tone Difference Matrix** | | | |
| Coarseness | 0,806 [2] | 0,526 [2] | 0,013 [2] |
| Contrast | 0,053 [2] | 0,219 [2] | 1,000 [1] |
| Busyness | 0,287 [2] | 0,468 [2] | 0,008 [2] |
| Complexity | 0,002 [2] | 1,000 [1] | 0,001 [2] |
| Strength | 0,575 [2] | 0,394 [2] | 0,044 [2] |

[*] Atypical: Atypical pneumonia group.

1. Kruskal Wallis and post-hoc Mann Whitney test.

2. T-test.

GLNUN: Gray Level Non-uniformity Normalized, RLNUN: Run Length Non-uniformity Normalized, LGLE: Low Gray Level Emphasis, HGLE: High Gray Level Emphasis, SZNUN: Size Zone Non-uniformity Normalized, DNUN: Dependency Non-uniformity Normalized, IDMN: Inverse Difference Moment Normalized, IDN: Inverse Difference Normalized.

tended to classify peripheral lesions as COVID-19 lesions and nonperipheral lesions as AP lesions more often. In our study, Model-4 correctly classified 48.6% peripheral and 87.8% non-peripheral AP lesions and 86.5% peripheral and 36.8% nonperipheral COVID-19 lesions.

The inclusion of Sphericity and lesion location to build a four-parameter model yielded a model with slightly lower sensitivity, specificity and accuracy (Model-5, Table 6). This model correctly predicted 48.6% peripherally located AP lesions, similar to Model-4 (case-by-case estimates were not exactly the same), and 82.7% peripheral COVID-19 lesions.

The best 4-parameter model according to the BIC analysis included the Interquartile range and Lesion location parameters (Model-6, Table 6). This model had the same sensitivity with Model-3 (case-by-case estimates were not exactly the same) and the second-best specificity and accuracy after Model-3. The VIF for Interquartile range and Range was calculated as 1.006.

Five out of 7 consolidations with wide ground-glass halos (5 AP and 2 COVID-19) were correctly predicted and the same 1 AP and 1 COVID-19 lesions could not be correctly predicted by all of the positive skewness models.

As a result, the best accuracy was achieved with Model-3, and the score for a lesion was calculated as:

$$\text{PSG Score} = \frac{1}{1 + e^{-(12,617 + 3,267 * log_{10}(Sphericity) - 7,497 * log_{10}(Range) - 5,918 * log_{10}(GLCM-Contrast))}} \quad (2)$$

The NSG included 56 (56%) AP lesions in 39 patients and 44 (44%) COVID-19 lesions in 29 patients. The NSG was primarily composed of consolidations (92 lesions). There were 8 GGO

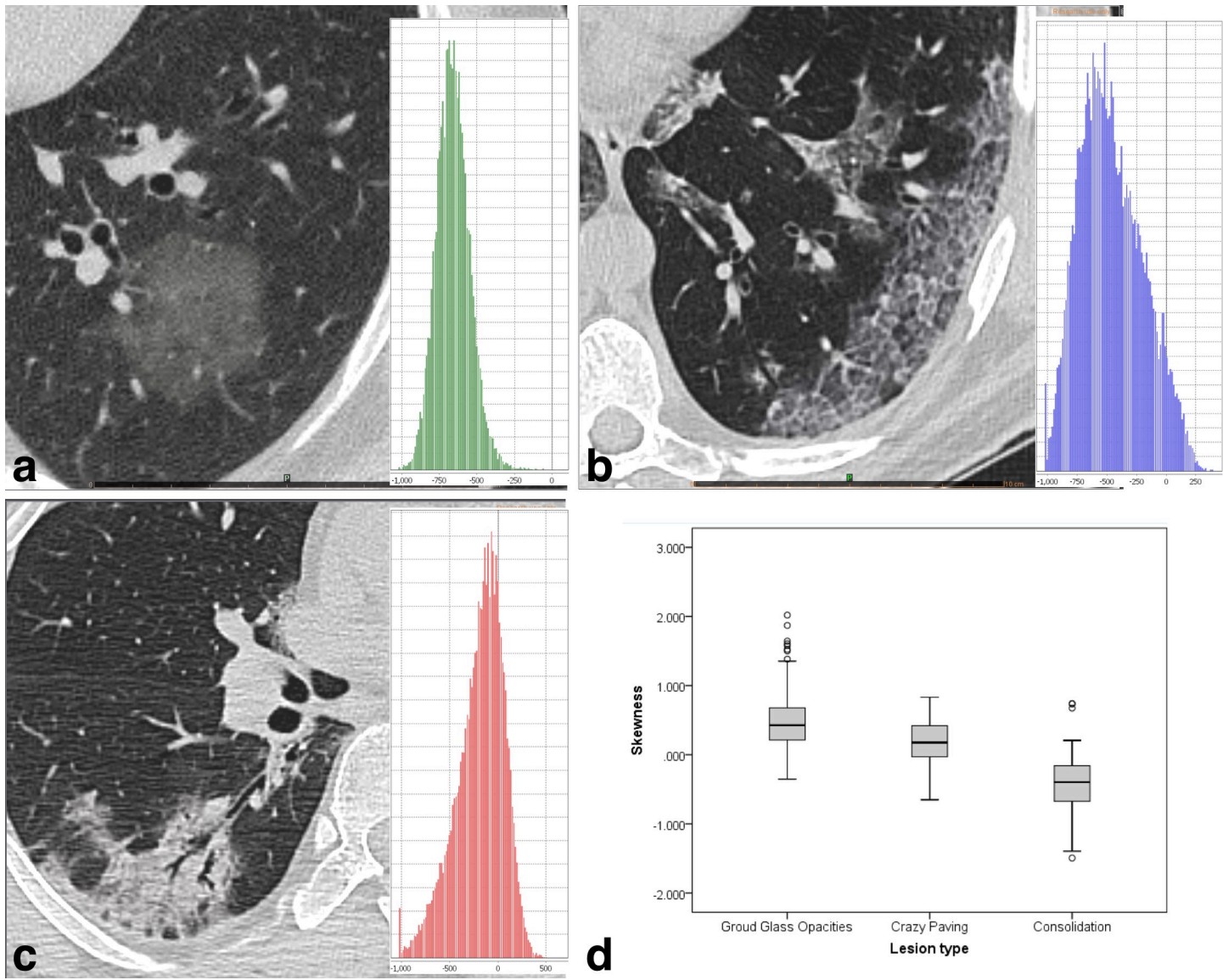

**Fig 2. Frequency distribution plot (histogram) of COVID-19-related lesions.** (a) ground glass opacity, (b) crazy paving, and (c) consolidation. Note that the histogram of the consolidation is right-skewed and (d) mean skewness value is negative.

lesions (7 COVID-19 and 1 AP) in the group, and their mean volume was 2.144 mL (1.286–2.773 mL).

The models for NSG prediction did not include Shape or Location parameters. Two texture parameters, GLCM-Inverse Difference Normalized (IDN) and the First-order parameter Mean Absolute Deviation (MAD), formed the basis of NSG estimation.

In the BIC analysis, the lowest-scoring 3-parameter model was formed by adding Spherical Disproportion to the above 2-parameter model (Model-7, Table 6) and the lowest-scoring 4-parameter model was formed by adding the Flatness parameter to this 3-parameter model (Model-8, Table 6). However, IDN and MAD were the only statistically significant parameters in the models. When the Spherical Disproportion parameter was changed with Sphericity ($p = 0.072$) or Lesion location ($p = 0.386$), these parameters were also unable to create a

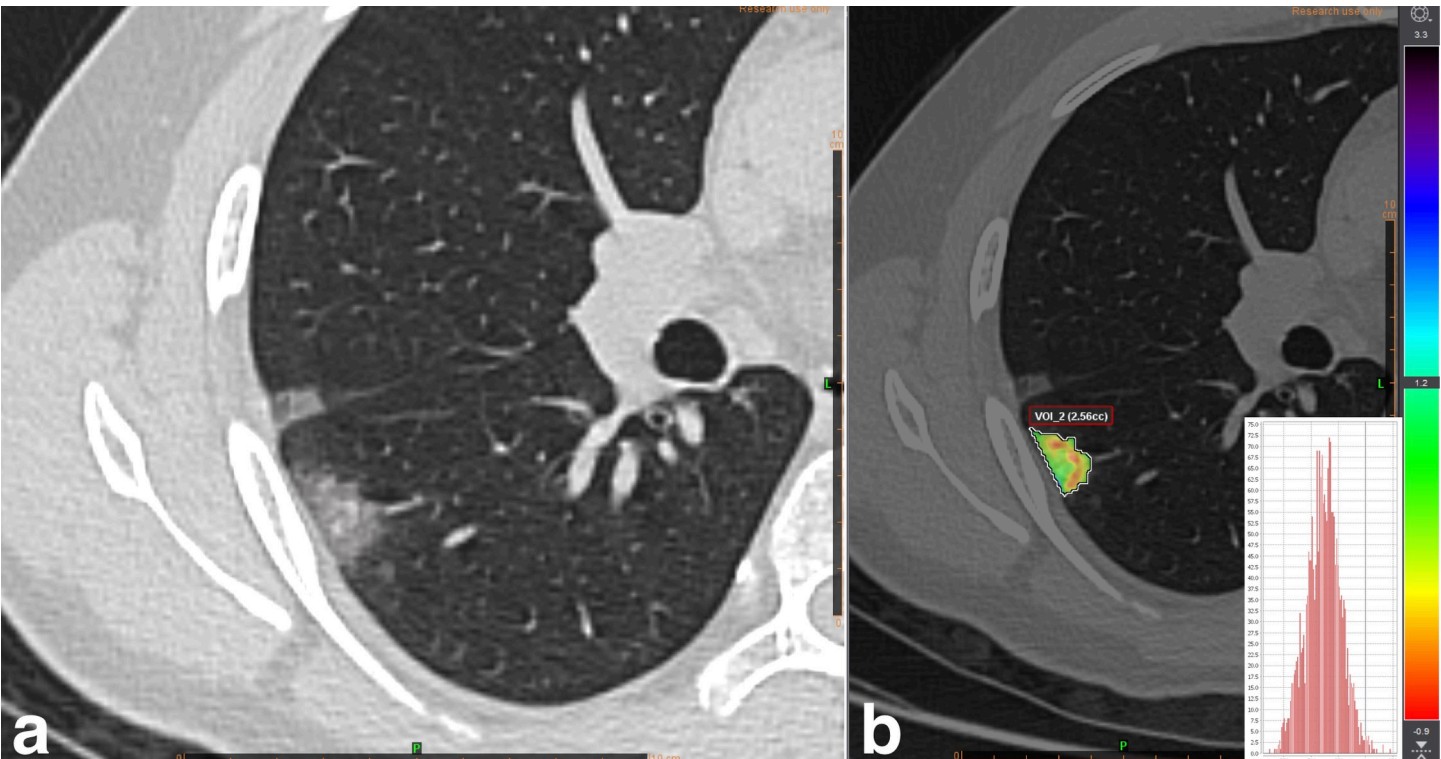

**Fig 3. GGO smaller than 3 mL.** (a) Gray scale CT image and (b) Skewness map of a 2.6 mL GGO lesion showed that high density septal components lead a right-shift.

statistically significant covariate. Moreover, in the cross-validation study, the case-by-case results of Model 7 and Model 8 were exactly the same.

Accordingly, the NSG score for Model-7 is depicted in Eq 3:

$$\text{NSG Score} = \frac{1}{1 + e^{-(-24,338-(5,290*log_{10}(Spherical\ Disproportion)+18,715*log_{10}(MAD)+276,037*log_{10}(IDN))}} \qquad (3)$$

Five out of 8 GGO lesions smaller than 3 mL that produced negative skewness values were correctly predicted by the NSG models. On the other hand, Model 3 of PSG was able to accurately predict all lesions. The Sphericity of the AP GGO lesion (from a patient diagnosed with RSV) was 0.362, while the COVID-19-related GGO lesions had a Sphericity between 0.550–0.669.

The net benefit provided by the highest accuracy PSG and NSG models (Model-3 and Model-7, respectively) was evaluated with decision curve analysis, using all possible threshold probabilities. While the PSG model did not differ from an approach in which all lesions were evaluated as COVID-19 for low threshold probabilities, it had a higher net benefit for the intermediate and high threshold (0.21–0.82) probability range (Fig 4A). On the other hand, the NSG model provided higher net benefit at all threshold probabilities (Fig 4B).

### Case-by-case evaluation

The results of Model-3 and Model-7, which had the highest accuracies in the cross-validation, were evaluated on a case-by-case basis. Regardless of the number of segmentations, the model was considered unsuccessful for a patient who had one falsely predicted lesion.

**Table 6. Model comparison in COVID-19 and atypical pneumonia prediction.**

| No | Parameters | p value | Odds Ratio*(95% CI) | AUC | Specificity % | Sensitivity % | Accuracy % | Specificity % | Sensitivity % | Accuracy % |
|---|---|---|---|---|---|---|---|---|---|---|
| | **Model Features** | | | | **Training Set** | | | **Test Set** | | |
| **GGO & consolidation** | | | | | | | | | | |
| **1** | Standard deviation | 0.000 | 0.446 (0.320–0.620) | 0.778 | 71.6 | 71.3 | 71.5 | 73.9 | 70.4 | 72.3 |
| | Spherical disproportion | 0.005 | 0.643 (0.471–0.878) | | | | | | | |
| | Lesion location | 0.000 | 2.802 (1.578–4.977) | | | | | | | |
| **2** | GLCM Contrast | 0.002 | 0.087 (0.019–0.402) | 0.797 | 71.6 | 75.7 | 73.5 | 67.2 | 73.0 | 69.9 |
| | Difference Average | 0.004 | 8.426 (1.991–35.668) | | | | | | | |
| | Range | 0.000 | 0.457 (0.325–0.642) | | | | | | | |
| | Lesion location | 0.000 | 2.922 (1.629–5.242) | | | | | | | |
| **PSG** | | | | | | | | | | |
| **3** | GLCM Contrast | 0,000 | 0.372 (0.216–0.641) | 0.774 | 76.9 | 77.5 | 77.2 | 76.9 | 74.6 | 75.8 |
| | Range | 0,005 | 0.043 (0.005–0.382) | | | | | | | |
| | Sphericity | 0,038 | 1.500 (1.004–2.240) | | | | | | | |
| **4** | GLCM Contrast | 0.001 | 0.417 (0.249–0.699) | 0.795 | 69.2 | 74.7 | 71.8 | 69.2 | 73.2 | 71.1 |
| | Range | 0.001 | 0.4025 (0.003–0.230) | | | | | | | |
| | Lesion location | 0.003 | 3.046 (1.491–6.898) | | | | | | | |
| **5** | GLCM Contrast | 0,000 | 0.378 (0.220–0.650) | 0.802 | 71.8 | 74.7 | 73.2 | 67.9 | 71.8 | 69.8 |
| | Range | 0,009 | 0.045 (0.004–0.452) | | | | | | | |
| | Sphericity | 0,038 | 1.553 (1.025–2.352) | | | | | | | |
| | Lesion location | 0,005 | 2.993 (1.385–6.465) | | | | | | | |
| **6** | GLCM Contrast | 0.000 | 0.204 (0.086–0.485) | 0.806 | 73.1 | 76.1 | 74.5 | 70.5 | 74.6 | 72.5 |
| | Range | 0.000 | 0.323 (0.175–0.598) | | | | | | | |
| | Interquartile Range | 0.033 | 2.431 (1.073–5.510) | | | | | | | |
| | Lesion location | 0.002 | 3.365 (1.539–7.356) | | | | | | | |
| **NSG** | | | | | | | | | | |
| **7** | IDN | 0.000 | 0.183 (0.083–0.406) | 0.907 | 87.5 | 79.5 | 84.0 | 85.7 | 79.5 | 83.0 |
| | Mean absolute deviation | 0.000 | 0.114 (0.035–0.364) | | | | | | | |
| | Spherical disproportion | 0.124 | 0.591 (0.302–1.155) | | | | | | | |
| **8** | IDN | 0.000 | 0.186 (0.084–0.410) | 0.910 | 85.7 | 87.1 | 85.0 | 85.7 | 79.5 | 83.0 |
| | Mean absolute deviation | 0.000 | 0.117 (0.037–0.373) | | | | | | | |
| | Spherical disproportion | 0.174 | 0.623 (0.315–1.233) | | | | | | | |
| | Flatness | 0.304 | 1.384 (0.744–2.571) | | | | | | | |

* Odds ratio values were obtained using standardized parameters to allow comparison. GGO: Ground-glass opacities, PSG: Positive skewness group, NSG: Negative skewness group, GLCM: Gray Level Co-occurrence Matrix, IDN: Inverse Difference Normalized.

In the PSG, 10 AP (2 single and 8 multiple lesions) and 13 COVID-19 (10 single and 3 multiple lesions) patients were falsely predicted by Model-3. The remaining 39 AP (79.6%) and 24 COVID-19 patients (64.9%) were correctly predicted.

In NSG, 6 AP (4 single and 2 multiple lesions) and 4 COVID-19 (2 single and 2 multiple lesions) patients were falsely predicted by Model-7. The remaining 33 AP patients (84.6%) and 25 COVID-19 patients (86.2%) were correctly predicted.

Some exchanges were made between the groups. A total of 20 patients (14 AP, 6 COVID-19) had both NSG and PSG lesions at the same time. Five of these patients (1 AP, 4 COVID-19) were falsely predicted by the PSG model, while the NSG model correctly predicted them all.

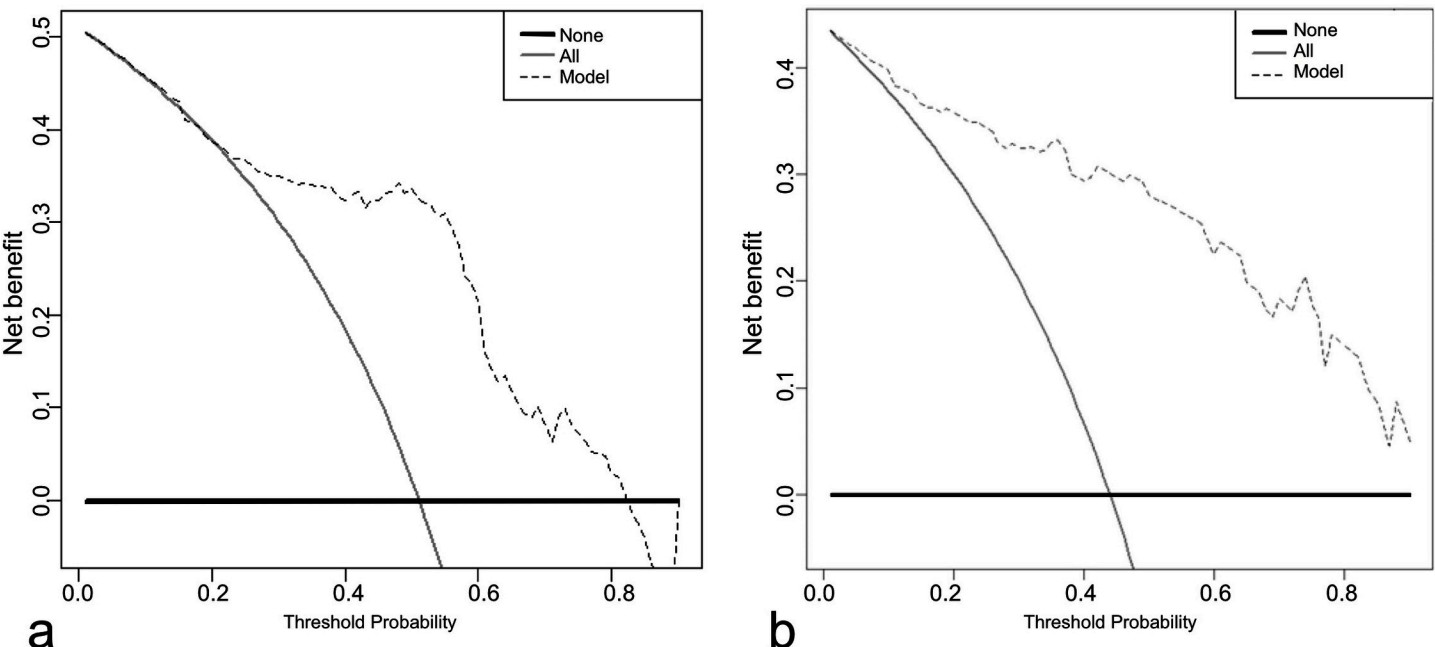

**Fig 4. Decision curve analysis of the models with highest accuracy.** (a) PSG Model-3, and (b) NSG Model-7 None (thick line): Net benefit if all patients were accepted as atypical pneumonia, All (thin line): Net benefit if all patients were accepted as COVID-19, Model (dashed line): Net benefit if patients were managed according to model. Note that PSG model was not superior to manage all patients as COVID-19 in lower threshold probabilities.

In contrast, 1 AP and 4 COVID-19 patients presented with a total of 8 GGO lesions with a volume less than 3 mL. The NSG model incorrectly predicted the AP patient and 1 COVID-19 patient, while PSG correctly predicted them all.

According to these results, we reached two basic rules and a simple algorithm for patient evaluation: (1) If a patient has both PSG and NSG lesions, the prediction should be made using NSG lesion(s) and NSG model; and (2) A lesion with a volume of less than 3 mL should be evaluated with the PSG model. Using an algorithm based on these rules, our final accuracy was 80% for COVID-19 and 81.1% for AP (Fig 5).

## Power analysis

An a priori power analysis was performed for independent samples t tests to determine the sample size of the COVID-19 and AP groups. The parameters used for this purpose were two tails, Cohen's d = 0.5, alpha = 0.05 and targeted power = 0.80, and the total sample size was calculated as 128.

During the study, the COVID-19 group consisted of 60 cases and the AP group consisted of 74 cases (total n = 134) and the power was calculated as 0.82 with post hoc analysis.

## Discussion

A total of 89 radiomic parameters, including Lesion location, Size, Shape, First-order and Second-order texture parameters, were evaluated, and none could individually differentiate COVID-19 from AP with sufficient sensitivity and specificity. For this reason, a method based on the estimation of lesions was adopted by creating models with logistic regression analysis. Although 24 parameters were significantly different between COVID-19 and AP-related GGO and consolidations, none of the parameters, except Range, could differentiate CP lesions

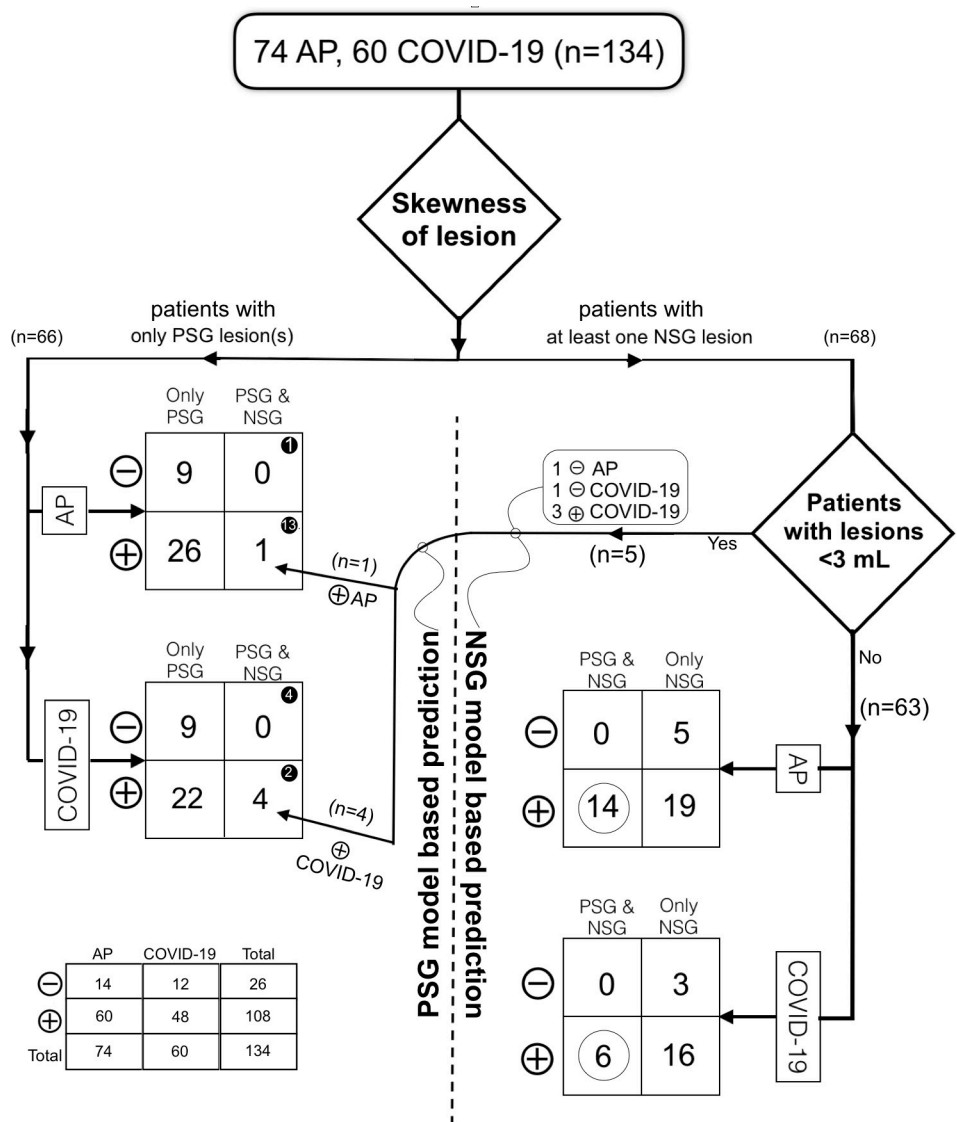

**Fig 5. An algorithm to predict pneumonic lesions with dedicated logistic regression models.** AP Atypical pneumonia, PSG Positive skewness group, NSG Negative skewness group, Number of, ⊖ Falsely predicted patients, and ⊕ Correctly predicted patients. Note the distribution of 20 patients with both types of lesions that black circled in PSG and white circled in NSG model predictions.

between the two disease groups. Thus, we focused on creating models that predicted GGOs and consolidations.

In our study, a one-model-for-all-lesions approach resulted in false estimates in the GGO and consolidation subgroups. Fang et al. reported a single model that could differentiate COVID-19 and influenza with high AUCs [22]. However, their model was not composed of radiomic parameters alone but also included parameters such as mediastinal lymphadenopathy and pleural effusion that are rarely reported for COVID-19 [2].

The models that predicted PSG (predominantly GGO) lesions always included the parameters GLCM-Contrast and Range. GLCM-Contrast represents the local gray-level variations within the lesion, and wrinkled images or images with edges have high values [23]. Range is the difference between the highest and lowest voxel densities in the lesion. As the values of

these two parameters increased, the possibility of the model classifying the lesion as an AP lesion also increased. The incorporation of these two parameters allowed the models to predict round or peripheral AP lesions with good specificity. The Expert Consensus Statement on COVID-19 reporting describes a typical COVID-19 lesion as a peripheral, bilateral, round GGO lesion [2]. Our study results for GGO lesions are consistent with the statement, as we found that Sphericity and Lesion location were prominent parameters. In the training and validation sets, the model with the highest accuracy contained the Sphericity parameter. On the other hand, models with standardized parameters showed that the Lesion location parameter had the highest odds ratio. Coronaviruses other than SARS-CoV-2 and influenza virus lead to peripheral involvement more often than other AP viruses [2,24–26]. In our study, the proportion of peripheral lesions in the AP group (excluding CP lesions) was 47%, and the models with the Lesion location parameter misclassified AP lesions slightly more often than the model with the Sphericity parameter. Although successful on the positive class side (COVID-19), the low number of true negatives (AP) explains the low accuracy achieved in the location-based models despite their high AUCs.

In the NSG (predominantly consolidations), neither shape nor location had a significant effect on the models' performance. It has been reported that as COVID-19 progresses, round GGO lesions tend to evolve into patchy GGO lesions and consolidations [26]. Consolidation has been reported in up to 64% of influenza-related pneumonias [25,27], and lobar and segmental consolidations are known to develop in pulmonary infections associated with influenza virus, adenovirus and human coronaviruses other than SARS-CoV-2 [28]. The NSG models were based on the parameters IDN and MAD. IDN measures local homogeneity, and larger values indicate a more homogeneous texture on a local scale [23]. The parameter MAD measures the distribution of voxels, and after the parameters SD, Range and IR, another parameter evaluating the voxel distribution was included in a model. It was found that despite the wider gray scale distribution of the voxels of the AP-related consolidations, as we also found for the GGOs, their local homogeneities were also greater than those of COVID-19-related consolidations.

Positively skewed COVID-19 GGO lesions with a volume of less than 3 mL were distinctly spherical, while one AP GGO lesion had a low sphericity. Thus, while NSG models were not successful for small volume lesions, Model 3 accurately predicted all lesions. Although spherical lesions in measles and varicella-zoster virus-related pneumonia have already been described [28], the shape-related features of small AP lesions should be investigated in future studies.

In this study, no parameter other than a radiomic feature was included in the models. Additionally, AP patients with tree-in-bud and pleural effusion were not included in the study; thus, the discriminability between same-category COVID-19 pneumonia and AP-associated lesions was investigated. The NSG models, which consisted mostly of consolidations, showed higher accuracy than the PSG models, which included mainly GGO lesions. It was seen that a higher net benefit could be obtained through our models according to a theoretical condition in which all lesions were evaluated as COVID-19 or AP lesions. Moreover, with the algorithm described in our study, an accuracy of 80% was achieved for both the COVID-19 pneumonia and AP groups without using any data other than radiomic parameters.

Reproducibility is the main problem of radiomics studies [16]. Although there are suggested methods to compensate for device and protocol-related differences [29], the images were obtained from the same device and protocol in our study. Additionally, the voxel densities were implemented as normalized values, not directly as Hounsfield units. Although some software can discriminate healthy and diseased parenchymal areas at the lung scale for processing all individual lesions as one large composite lesion [20], predicting different lesion types with a single model led to false negativity in our study. The lesions were manually segmented with

simple rules, and we showed that there was no significant difference between the segmentations of the different observers.

This study has some limitations. First, there were a few serologically diagnosed AP patients in the retrospective screening. Additionally, the mean number of lesions detected per patient was lower than that among COVID-19 patients. Since it is necessary to work with balanced groups to demonstrate the effectiveness of the models, the largest possible AP group was created, and then the number of COVID-19 patients required was determined according to the results of a power analysis. Thus, our sample size was relatively small. Second, the time between the onset of symptoms and the CT scan was slightly longer in the AP group, and the number of consolidations recorded in this group was also higher. Since each patient's CT examination obtained prior to antiviral treatment was included in the study, any follow-up films were not used. Finally, an effective model for CP lesions could not be developed with the methods that we used to calculate the radiomics features. In the future, we aim to develop efficient models by using series containing more AP-associated CP lesions and different calculation methods.

In conclusion, using lesion-dedicated models consisting of only radiomics parameters and an algorithm that combined the appropriate lesion type for the correct model, we showed that COVID-19- and AP-associated GGO lesions and consolidations could be predicted with good accuracy. Our validation studies showed that roundness and peripheral location were the strongest parameters for associating a GGO lesion with COVID-19, although both were found ineffective in predicting a lesion in the consolidation stage.

## Author Contributions

**Conceptualization:** Mutlu Gülbay.

**Data curation:** Mutlu Gülbay, Bahadır Orkun Özbay, Bökebatur Ahmet Raşit Mendi, Aliye Baştuğ, Hürrem Bodur.

**Formal analysis:** Mutlu Gülbay, Aliye Baştuğ, Hürrem Bodur.

**Funding acquisition:** Mutlu Gülbay.

**Investigation:** Mutlu Gülbay, Bahadır Orkun Özbay, Bökebatur Ahmet Raşit Mendi.

**Methodology:** Mutlu Gülbay, Bökebatur Ahmet Raşit Mendi, Hürrem Bodur.

**Project administration:** Mutlu Gülbay, Bahadır Orkun Özbay, Bökebatur Ahmet Raşit Mendi.

**Supervision:** Mutlu Gülbay, Bökebatur Ahmet Raşit Mendi, Aliye Baştuğ, Hürrem Bodur.

**Validation:** Mutlu Gülbay, Bökebatur Ahmet Raşit Mendi.

**Writing – original draft:** Mutlu Gülbay.

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
