## [Decision Letter · Decision Letter 0]

22 Jan 2021

A CT Radiomics Analysis of COVID-19-related Ground-Glass Opacities and Consolidation: Is It Valuable in a Differential Diagnosis with Other Atypical Pneumonias?

PONE-D-20-34364

Dear Dr. Gulbay,

We’re pleased to inform you that your manuscript has been judged scientifically suitable for publication and will be formally accepted for publication once it meets all outstanding technical requirements and the comments of the Academic Editor to the Authors. Please read the comments of the Reviewers and include the proposed changes in the correction of the proof that will be sent to you.

Kind regards,

Domokos Máthé

Academic Editor

PLOS ONE

Additional Editor Comments (optional):

This is a manuscript and a well handled topic, meriting immediate acceptance, therefore I opted to issue the acceptance letter based on the comments of the reviewers. However, I request the Authors to correct the proof in a way that they include a section about data normality testing and the scientific general basis of lung / lung-lesion selection reasons.

Reviewers' comments:

Reviewer's Responses to Questions

**Comments to the Author**

1. Is the manuscript technically sound, and do the data support the conclusions?

Reviewer #1: Yes

Reviewer #2: Yes

2. Has the statistical analysis been performed appropriately and rigorously? 

Reviewer #1: Yes

Reviewer #2: Yes

3. Have the authors made all data underlying the findings in their manuscript fully available?

Reviewer #1: Yes

Reviewer #2: Yes

4. Is the manuscript presented in an intelligible fashion and written in standard English?

Reviewer #1: Yes

Reviewer #2: Yes

5. Review Comments to the Author

Reviewer #1: It would be benefitial if the intro included a highlight of what are the typical use cases of radiology-only discovery of covid. In your case it would be asymptotic covid patient accidentally diagnosed during other thorax examinations. For this reason I would also compare not to just AP, but other Ps that have lower rate of correct diagnosis clindically.

Reviewer #2: This manuscript is one of the gap-filling works that try to quantify the difference of lung in COVID and AP by radiomics methods. The main advantage of this work to precisely experimentally broadly validated the histomorphology and CT image of the lung.

(My Academic Editor`s edits to the comments of this reviewer are in marker.)

1. It is not clear to me how the patients of AP groups were selected. Please include a sentence on selection.

2. The least documented part of this work the selection methods of the lesions. I could not understand how the full lung status of any given patient is characterized by 1-4 lesions. How do you select the characteristic, defining lesions form the large available quantity of each patient`s lung lesions. Please include a section on this.

3. The authors used a big amount of statistical methods to evaluate the results. My first question would be how to determine the number of patients? Do you think this amount is enough for final conclusion? I suppose you have tested the normality of the data before you selected the parametric and nonparametric statistical methods. Could you show the results of these normality tests? Please include the appropriate details there.

6. PLOS authors have the option to publish the peer review history of their article (what does this mean?). If published, this will include your full peer review and any attached files.

Reviewer #1: No

Reviewer #2: No

---

## [Editor Report · Acceptance letter]

25 Feb 2021

PONE-D-20-34364 

A CT Radiomics Analysis of COVID-19-related Ground-Glass Opacities and Consolidation: Is It Valuable in a Differential Diagnosis with Other Atypical Pneumonias? 

Dear Dr. Gülbay:

I'm pleased to inform you that your manuscript has been deemed suitable for publication in PLOS ONE. Congratulations! Your manuscript is now with our production department. 

Kind regards, 

on behalf of

Dr. Domokos Máthé 

Academic Editor

PLOS ONE